# Enhancing Multimodal Product Retrieval in E-Commerce by Reversing Typographic Attacks

**Janet Jenq and Hongda Shen**[*]
PitchBook
{janet.jenq, hongda.shen}@pitchbook.com

## Abstract

Multimodal product retrieval systems in e-commerce platforms rely on effectively combining visual and textual signals to improve search relevance and user experience. However, vision-language models such as CLIP are vulnerable to typographic attacks, where misleading or irrelevant text embedded in images skews model predictions. In this work, we propose a novel method that reverses the logic of typographic attacks by overlaying relevant textual content (e.g., titles, descriptions) directly onto product images to perform additional vision-text compression, thereby strengthening image-text alignment and boosting multimodal product retrieval performance. We evaluate our method on three vertical-specific e-commerce datasets (sneakers, handbags, and trading cards) using five state-of-the-art vision foundation models. Our experiments demonstrate consistent improvements in unimodal and multimodal retrieval accuracy across categories and model families. Our findings suggest that visually rendering product metadata is a simple yet effective enhancement for zero-shot multimodal retrieval in e-commerce applications.

## 1 Introduction

In e-commerce platforms, buyers and sellers engage with product listings through visual and textual information by viewing images, reading titles and descriptions, or searching with image- and keyword-based queries Ren et al. (2018); Liu and Ramos (2025). Multimodal product retrieval matches queries to relevant items by combining signals from image and text when either partial (e.g., only an image or a title) information or full (both image and title) information is available. By incorporating signals from both modalities, retrieval models can better capture user intent and improve result relevance. In practice, state-of-the-art models such as CLIP are widely adopted because their dual-encoder architecture, which encodes text and image inputs into a shared embedding space, allows efficient nearest-neighbor retrieval across unimodal and multimodal scenarios. The scalability and zero-shot generalization of these multimodal models make them particularly suited for large-scale e-commerce applications with vast search space and query diversity.

An emerging line of research has focused on understanding the impact of rendered visual text within images on the representations learned by contrastive vision-language models, particularly those in the CLIP family Goh et al. (2021); Noever and Noever (2021); Lemesle et al. (2022); Qraitem et al. (2025a); Cheng et al. (2025a); Qraitem et al. (2025b). Earlier studies Goh et al. (2021); Lemesle et al. (2022); Qraitem et al. (2025b) have demonstrated that overlaying irrelevant or misleading text on images can act as a form of adversarial attack against multimodal representation models. Specifically, these works introduce the notion of *typographic attacks*, in which visually rendered textual content manipulates model predictions by injecting misleading semantics. Large Vision-Language Models such as the CLIP family are trained on large-scale, weakly curated web data, where image captions often describe visible elements rather than highlighting features that are most

---

[*]Equal contribution. Work performed while the authors were at eBay.

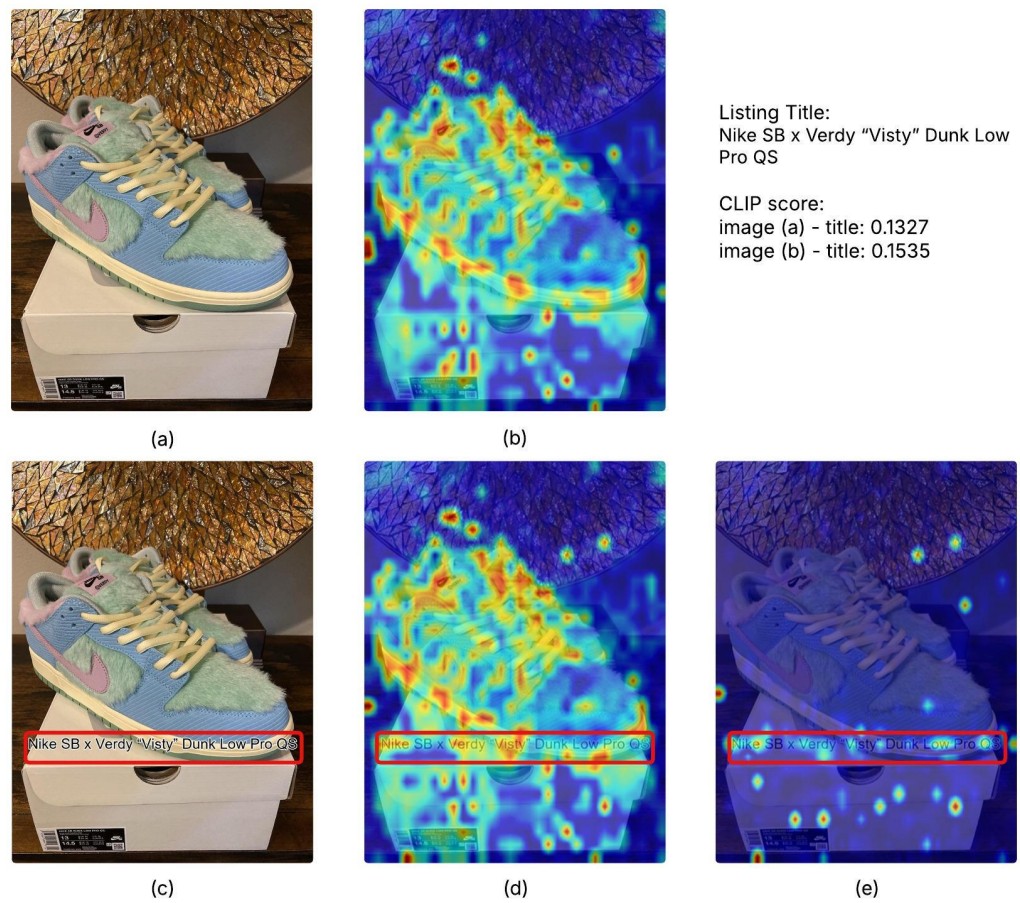

Listing Title:
Nike SB x Verdy "Visty" Dunk Low Pro QS

CLIP score:
image (a) - title: 0.1327
image (b) - title: 0.1535

Figure 1: Visualization of the impact of title rendering on image representations, shown through changes in the attention sensitivity heatmap for the OpenAI CLIP ViT-Large model.

relevant for understanding the image. As a result, these models tend to attend to spurious correlations, which are patterns that emerge when visual and textual elements frequently appear together in the training data despite having no meaningful semantic relationship Qraitem et al. (2025b). In a related but distinct line of work, CLIPPO Tschannen et al. (2023) explores the possibility of building a unified representation for both image and text modalities by rendering text as image and processing both through a shared visual encoder. Their approach eliminates the need for modality-specific components, thereby simplifying the architecture and reducing the parameter count by approximately 50% compared to dual-encoder models such as CLIP.

Recently, DeepSeek-OCR Wei et al. (2025) introduced a vision–text compression paradigm that renders textual content onto images, enabling visual encoders to absorb linguistic information and thereby unify vision and language representations for efficient OCR. Glyph Cheng et al. (2025b), released shortly thereafter, generalized this concept to long-context modeling by rendering extended text sequences into visual forms processed by a VLM, achieving scalable context compression while maintaining semantic fidelity.

Motivated by these advances in vision–text compression, we propose a novel approach for enhancing multimodal product retrieval by reversing the logic of typographic attacks. Rather than injecting adversarial or misleading text, our method renders product-relevant information—such as titles, descriptions, and metadata—directly onto product images, thereby strengthening image–text alignment and improving retrieval performance. Unlike CLIPPO, which renders textual content into a separate image input, our approach embeds the text directly onto the original product image. This

consolidation reduces the number of image inputs from two to one, reducing computational overhead during inference. An example of our method is shown in Figure 1, using the CLIP ViT-Large model. Subplot (a) shows a product image (Nike sneakers), and (c) shows the same image with the listing title rendered on top using our algorithm. Subplots (b) and (d) display attention sensitivity heatmaps generated using the method of Bousselham et al. (2025) for the same text query applied to the raw and title-rendered images, respectively. The rendered title shifts attention toward the text region (highlighted by the red box), with Subplot (d) showing increased focus on semantic entities like "Nike" and "Verdy's Visty". In Subplot (e), we visualize the absolute difference between the two heatmaps, where changes are concentrated near the rendered title area. This attention shift suggests that title-rendered images yield embeddings more aligned with their textual counterparts, improving multimodal retrieval. In this example, the CLIP similarity score between the image and its title increases from 0.1328 (raw image) to 0.1535 (title-rendered image), indicating improved image-text alignment.

## 2 Related Work

Recent advances in vision-language pre-training demonstrate the effectiveness of integrating textual and visual information for e-commerce product retrieval applications Zheng et al. (2023a,b); Yu et al. (2022). In particular, Yu et al. (2022) introduced *CommerceMM*, a multimodal representation learning framework capable of handling a wide range of e-commerce tasks, including multimodal categorization, image-text retrieval, query-to-product retrieval, and image-to-product retrieval. To improve product matching in the fashion domain, Tóth et al. (2024) proposed a model that integrates multiple image encoders and a single text encoder. However, the lack of open-source implementations and the domain-specific nature of these approaches restrict their utility in broader multimodal product retrieval settings.

CLIP Radford et al. (2021) established a new paradigm in visual representation learning by employing natural language supervision. Building upon its success, several variants such as SigLIP Zhai et al. (2023) and SigLIP 2 Tschannen et al. (2025) have been developed to further advance multimodal representation learning. Additionally, the Perception Encoder (PE) Bolya et al. (2025) family of vision encoders has achieved state-of-the-art performance across numerous vision tasks. By combining a robust contrastive pretraining objective with fine-tuning on synthetically aligned videos, PE models produce generalizable and scalable features for downstream tasks.

To the best of our knowledge, the term *typographic attack* was first introduced in Goh et al. (2021) to describe a non-programmatic adversarial attack in which misleading or incorrect text is rendered directly onto an image to induce incorrect model predictions, particularly image classification tasks. Follow-up studies Jia et al. (2022); Azuma and Matsui (2023); Lemesle et al. (2022) have further investigated the mechanisms underlying typographic attacks and proposed various defense strategies. More recently, Qraitem et al. (2025b) extended the notion of typographic attacks to include non-matching text and graphical symbols, reflecting the prevalence of branding artifacts in web-scale training data. Although the root cause of such attacks remains underexplored, the prevailing view across the literature is that vision language models (VLMs), typically contrastive models like CLIP, are prone to learning spurious correlations, or unintended patterns arising from frequent but semantically unrelated co-occurrences Lin et al. (2025); Wang et al. (2024a). For instance, Lin et al. (2025) observed that embedded text and corresponding image captions can introduce a strong text-spotting bias, which is intrinsic to the contrastive learning framework. To mitigate this effect, they proposed a data filtering approach based on OCR tools. Subsequent work has confirmed this bias and introduced alternative filtering strategies for CLIP training Cao et al. (2023); Maini et al. (2024); Wang et al. (2024b). CLIP and SigLIP models are widely used as vision encoders in large multimodal models (LMMs), which combine vision-language models (VLMs) with large language models (LLMs) to support more advanced reasoning and interaction. Given the strong zero-shot performance of these encoders on tasks such as classification, retrieval, and segmentation, their sensitivity to typographic attacks is likely to propagate to LMMs as well Cheng et al. (2024, 2025c).

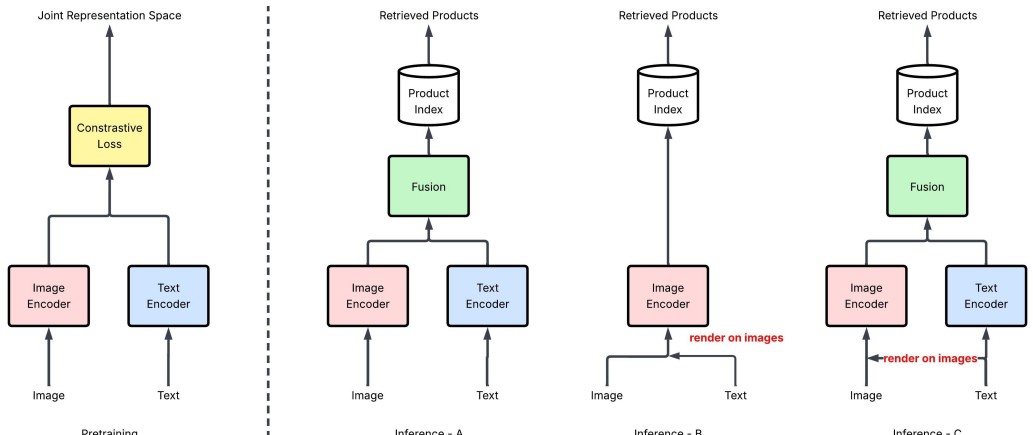

Figure 2: Overview of the proposed method, including two new inference modes that leverage text-rendered images for improved multimodal retrieval.

## 3 Methodology

### 3.1 Problem Formulation

We formulate multimodal product retrieval as the task of finding the most similar product given a query, where both the query and candidate products consist of an image and associated text. Given a query $q = (i_q, t_q)$, consisting of an image $i_q$ and its associated text $t_q$, the objective is to retrieve the product $p = (i_p, t_p)$ that most closely matches the query from a candidate pool of e-commerce products $\mathcal{P}$. Image and text inputs are independently encoded by an image encoder and a text encoder, and the resulting features are fused into a joint multimodal embedding space $f(i, t)$. Since text inputs can be lengthy (e.g., item titles, product descriptions, and attribute lists), we leverage a Large Language Model (LLM) to summarize them, preventing excessive textual content from overshadowing the image. This summarization also facilitates alignment with text encoders, particularly when input token limits are imposed. For simplicity, we refer to the summarized text as the *title* throughout this paper, while noting that such inputs may originate from multiple sources within an e-commerce listing. The similarity between the query and a candidate item is computed using a similarity function $S(f(i_q, t_q), f(i_p, t_p))$, where we use cosine similarity throughout this work. The best-matching product is retrieved by ranking the similarity scores between the query and each candidate product. The left and right subplots in Figure 2 illustrate the training procedure and the inference-time flow, respectively.

### 3.2 Our Method

Typographic attacks leverage unintended correlations between semantic concepts and unrelated visual signals to manipulate the representations learned via contrastive models such as CLIP. Instead of the misleading text, the proposed approach reverses it to render product metadata, e.g., title, on the image to strengthen image-visual text correlation which improves the product retrieval performance.

Figure 2 presents an overview of our method. The left subplot shows the pretraining stage, where a joint image-text embedding space is learned via contrastive learning. In zero-shot settings, models are trained on large-scale, weakly curated web data. These models can optionally be fine-tuned on domain-specific datasets when sufficient data is available.

The right subplot depicts three inference pathways for generating query and index embeddings. These embeddings are computed using a selected vision foundation model and can be indexed into a production-grade k-nearest neighbors (KNN) engine with tuning for performance, latency, and resource constraints. Inference-A shows standard multimodal retrieval, where image and text embeddings are fused. Inference-B and Inference-C, proposed in our method, incorporate text

rendering onto the image before generating the corresponding embeddings. Inference-B uses the text-rendered image alone, effectively retaining multimodal input within a single visual stream. Denoting the text-rendered image as $\hat{i}$ and the image embedding function as $g(\cdot)$, retrieval reduces to finding the closest match via $S(g(\hat{i_q}), g(\hat{i_p}))$. Inference-C extends this one-encoder design by fusing the embedding of the text-rendered image with a separately encoded text embedding, thereby forming a hybrid multimodal representation. The similarity function is accordingly modified to $S(f(\hat{i_q}, t_q), f(\hat{i_p}, t_p))$. **Note that text rendering is applied only during embedding generation and does not alter or replace the original images, thus avoiding any aesthetic or user-facing impact.**

---

**Algorithm 1** Render Scaled Centered Text on an Image

---

**Require:** image $I$ of width $W$ and height $H$, text to render $T$
**Ensure:** Draw text $T$ centered on the image $I$
 1: **function** GETMAXFONTSIZE($I, T$)
 2:     W, H = size(I)
 3:     $s_{\max}$, $s_{\min}$, $w_{\max} \leftarrow \lfloor W/6 \rfloor, 1, 0.9 \cdot W$
 4:     **while** $s_{\min} < s_{\max}$ **do**
 5:         $s_{\mathrm{mid}} \leftarrow \lfloor (s_{\min} + s_{\max} + 1)/2 \rfloor$
 6:         Load font with size $s_{\mathrm{mid}}$
 7:         Measure text width $w_{\mathrm{mid}}$
 8:         **if** $w_{\mathrm{mid}} \leq w_{\max}$ **then**
 9:             $s_{\min} \leftarrow s_{\mathrm{mid}}$
10:         **else**
11:             $s_{\max} \leftarrow s_{\mathrm{mid}} - 1$
12:         **end if**
13:     **end while**
14:     **return** $s_{\min}$
15: **end function**
16: **function** RENDERTEXT($I, T$)
17:     W, H = size(I)
18:     $s \leftarrow$ GETMAXFONTSIZE($I, T$)
19:     Load font with size $s$
20:     Measure text dimensions $(w_T, h_T)$
21:     $x \leftarrow 0.5 \cdot W - 0.5 \cdot w_T$
22:     $y \leftarrow 0.5 \cdot H - 0.5 \cdot h_T$
23:     Draw text $T$ at $(x, y)$ with font-size $s$
24: **end function**

---

To ensure the rendered text remains legible across diverse inputs, we implement a simple, computationally lightweight adaptive rendering algorithm that computes the maximum font size for a given text and centers it on the image using the `ImageDraw` module from PIL Clark et al. (2025). The pseudo-code for this text rendering procedure is provided in Algorithm 1. Although Algorithm 1 employs a fixed font style and an adaptive font-size selection method for visual clarity, the variations in rendering parameters, such as font type, font size, text color, or placement, may affect the quality of the learned embeddings and retrieval outcomes. A systematic evaluation of these design factors will be conducted in future work to provide further insights into the robustness and optimal configuration of the rendering process.

## 4   Experiments

### 4.1   Datasets and Metrics

We evaluate our method on three proprietary datasets from a major e-commerce platform, covering sneakers, handbags, and NBA trading cards. Dataset statistics are presented in Table 1. Each query has a single human-annotated ground-truth match.

NBA trading cards typically contain rich visual text (e.g., player and team names), while sneaker and handbag images often contain minimal text. This variation enables a controlled evaluation of how

Table 1: Statistics of datasets used in the experiments.

| Dataset | Query # | Product # |
|---|---|---|
| Sneakers | 1,986 | 139,567 |
| Handbags | 2,000 | 48,075 |
| Trading Cards | 1,127 | 400,683 |

| Model | Image | Text | Fusion | Sneakers | | Handbags | | Trading Cards | |
|---|---|---|---|---|---|---|---|---|---|
| | | | | Acc@1 | Acc@3 | Acc@1 | Acc@3 | Acc@1 | Acc@3 |
| CLIP | Raw | – | – | 0.2185 | 0.2578 | 0.0450 | 0.0600 | 0.1624 | 0.2396 |
| | Raw | Title | Sum | 0.7014 | 0.7578 | 0.2285 | 0.2900 | 0.5439 | 0.7276 |
| | Raw | Title | Concat | 0.7004 | 0.7563 | 0.2355 | 0.2950 | 0.5306 | 0.6983 |
| | Rendered | – | – | 0.2790 | 0.3132 | 0.0810 | 0.1195 | 0.2094 | 0.3132 |
| | Rendered | Title | Sum | **0.7175** | **0.7679** | **0.2515** | **0.3130** | **0.5972** | **0.7737** |
| | Rendered | Title | Concat | 0.6883 | 0.7356 | 0.2450 | 0.3045 | 0.5626 | 0.7329 |
| PE | Raw | – | – | 0.5272 | 0.5992 | 0.0870 | 0.1425 | 0.4694 | 0.6850 |
| | Raw | Title | Sum | 0.8036 | **0.8505** | 0.2435 | 0.3275 | 0.8128 | 0.9388 |
| | Raw | Title | Concat | 0.7925 | 0.8439 | 0.2545 | 0.3260 | 0.7720 | 0.9228 |
| | Rendered | – | – | 0.7180 | 0.7618 | 0.1820 | 0.2600 | 0.5484 | 0.7915 |
| | Rendered | Title | Sum | **0.8127** | 0.8484 | **0.2955** | 0.3635 | **0.8421** | **0.9414** |
| | Rendered | Title | Concat | 0.8006 | 0.8374 | 0.2940 | **0.3645** | 0.8030 | 0.9272 |
| SigLIP | Raw | – | – | 0.4547 | 0.5176 | 0.1245 | 0.2125 | 0.5590 | 0.7870 |
| | Raw | Title | Sum | 0.7684 | 0.8278 | 0.3035 | 0.3685 | 0.8350 | **0.9379** |
| | Raw | Title | Concat | 0.7523 | 0.8087 | 0.3075 | 0.3720 | 0.8163 | 0.9104 |
| | Rendered | – | – | 0.7523 | 0.7895 | 0.2910 | 0.3615 | 0.6761 | 0.8634 |
| | Rendered | Title | Sum | **0.8006** | **0.8429** | **0.3320** | **0.3930** | **0.8616** | 0.9335 |
| | Rendered | Title | Concat | 0.7835 | 0.8212 | 0.3295 | 0.3895 | 0.8270 | 0.9148 |
| SigLIP 2 | Raw | – | – | 0.4300 | 0.5000 | 0.1250 | 0.2075 | 0.4747 | 0.7249 |
| | Raw | Title | Sum | 0.6440 | **0.7407** | 0.0755 | 0.1240 | 0.2209 | 0.3372 |
| | Raw | Title | Concat | 0.6375 | 0.7402 | 0.0800 | 0.1300 | 0.2227 | 0.3407 |
| | Rendered | – | – | **0.6636** | 0.7105 | **0.2390** | **0.3200** | **0.5954** | **0.7533** |
| | Rendered | Title | Sum | 0.4607 | 0.5705 | 0.1275 | 0.1790 | 0.2422 | 0.3274 |
| | Rendered | Title | Concat | 0.4673 | 0.5801 | 0.1330 | 0.1835 | 0.2520 | 0.3319 |
| DINOv2 | Raw | – | – | **0.0358** | **0.0468** | 0.0645 | 0.1035 | 0.2760 | 0.4153 |
| | Rendered | – | – | 0.0352 | 0.0443 | **0.0695** | **0.1120** | **0.2964** | **0.4357** |

Table 2: Performance of candidate models across different image input types, retrieval modalities, and fusion strategies on three product categories. The highest metric value for each model and product category is highlighted in bold.

overlaying product metadata text affects multimodal retrieval performance. We report accuracy at rank k, $Acc@k$, defined as the proportion of queries for which the correct product is retrieved in the top k. We use $Acc@1$ and $Acc@3$ in our analysis.

Due to the maximum token length constraints of the text encoders in most pretrained models, we employ an LLM to summarize all available item information including titles, descriptions, and attribute lists into a title-length text whenever the full text input is too long for the text encoder. This pre-processing step ensures that the most salient product information is preserved while maintaining compatibility with text encoders. In addition, this design choice simplifies implementation, reduces the risk of truncation errors, and provides a consistent textual representation that can be seamlessly integrated into the proposed retrieval enhancement method.

## 4.2 Main Results

We evaluate our enhancement method on five state-of-the-art vision foundation models across three product categories. CLIP Radford et al. (2021), Perception Encoder (PE) Bolya et al. (2025), SigLIP Zhai et al. (2023), and SigLIP 2 Tschannen et al. (2025) are dual-encoder models trained with

contrastive learning to align image-text pairs in a shared embedding space. These models support zero-shot multimodal retrieval without requiring explicit labels or fine-grained annotations. SigLIP 2 extends the SigLIP design with self-distillation and masked prediction. We also include DINOv2, a vision-only model trained with self-supervised learning to assess whether our method benefits models trained without textual supervision. Since all CLIP-like models evaluated in this study are limited to a small number of input text tokens, we restrict the text modality to query/product titles. Longer textual fields such as descriptions or metadata frequently exceed this limit and risk being truncated, potentially leading to degraded semantic alignment.

Table 3: Overview of candidate models and their specifications.

| Collection | Model | Image Enc. | Text Enc. |
|---|---|---|---|
| OpenAI | CLIP-L/14 | ✓ | ✓ |
| PE | PE-Core-L14-336 | ✓ | ✓ |
| SigLIP | SigLIP-SO400M/384 | ✓ | ✓ |
| SigLIP 2 | SigLIP2-SO400M/384 | ✓ | ✓ |
| Meta (DINOv2) | DINOv2-Large | ✓ | – |

As shown in Table 2, all candidate models are tested with raw and title-rendered image inputs. The rendered input is generated by applying Algorithm 1, which overlays the title text onto the image. For multimodal retrieval, we evaluate two commonly used fusion strategies: *Sum* and *Concat*, resulting in six configurations per model. While other fusion strategies exist, identifying the optimal fusion method is beyond the scope of this work. All experiments are zero-shot with no fine-tuning to ensure a fair comparison. Model details are provided in Table 3.

From Table 2, we observe that across all three product categories, the highest performance metrics are generally obtained when models use title-rendered images, for both $Acc@1$ and $Acc@3$. This suggests that the proposed method consistently enhances product retrieval performance across all base models. Notably, SigLIP benefits the most, with an average improvement of 3.11 points in $Acc@1$ and 1.79 points in $Acc@3$ compared to the second-best configuration. This is particularly striking given that SigLIP already serves as the strongest baseline, indicating that the proposed enhancement can further improve even top-performing architectures. The Perception Encoder (PE) also demonstrates strong performance, showing comparable gains of 3.00 points in $Acc@1$ and 1.21 points in $Acc@3$, underscoring the general effectiveness of the proposed approach. Similar performance improvements are observed for the CLIP model as well.

SigLIP 2 exhibits some intriguing behavior in our experiments. When using raw images as input, multimodal retrieval with fused embeddings underperforms image-only retrieval in both the handbags and trading cards categories, whereas in the sneakers category, multimodal retrieval still provides a notable performance gain. Unlike CLIP, PE, and SigLIP, SigLIP 2 achieves its best results with title-rendered image-only retrieval. In particular, this configuration outperforms the second-best configuration by 8.21 points in $Acc@1$ and 3.69 points in $Acc@3$, suggesting that using a single image encoder on title-rendered inputs can serve as an effective alternative for multimodal retrieval when using SigLIP 2. One possible explanation is that the introduction of self-distillation and masked prediction in SigLIP 2's training may alter the alignment dynamics between image and text modalities in the joint embedding space learned during contrastive pretraining. However, we do not have definitive evidence to support this hypothesis, and further investigation is warranted.

From the empirical results in Table 2, we observe that the *Sum* fusion strategy consistently outperforms *Concat* across all models in the CLIP family. Furthermore, in all three categories, the proposed method using title-rendered images yields higher retrieval performance than raw image retrieval. It is also worth noting that the performance gains observed with DINOv2 are less significant compared to the contrastive learning-based models, likely due to the lack of multimodal pretraining in its architecture.

## 4.3 Case Study

Figure 1 presents an example featuring the Nike SB Dunk Pro Verdy Visty sneakers, which serves as the focus of this case study. Due to space constraints, we highlight results using the SigLIP model in this case study, which was selected based on its strong performance demonstrated in the previous section. We conducted two types of multimodal retrieval: (1) raw image combined with the title,

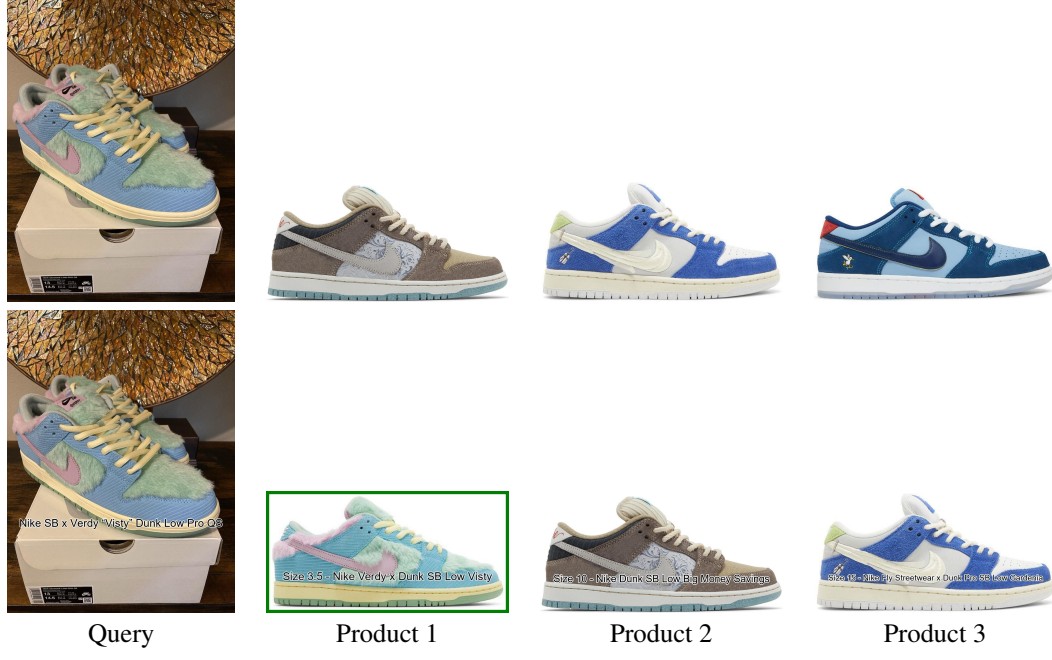

| Query | Product 1 | Product 2 | Product 3 |

Figure 3: Example from the sneakers category showing the query and retrieved products under raw (top row) and title-rendered (bottom row) conditions. The top row shows the original query image and three retrieved product images. The bottom row shows the corresponding images with listing titles rendered using our proposed method. The correct product match is highlighted with a green box.

and (2) title-rendered image combined with the title. Figure 3 shows the top-3 retrieved products for each setting. As shown, the proposed approach (bottom row), which uses the title-rendered image, successfully retrieves the correct product as the top-1 result. In contrast, the correct match is entirely missing from the top row, despite the visual similarity of other retrieved items. We have observed similar improvements using the proposed method across other samples and candidate models. Due to space constraints, additional qualitative examples of different categories are provided in the Appendix (Figures 4–10).

## 5 Conclusions and Future Work

In this paper, we introduce a novel approach for improving multimodal product retrieval by reversing the logic of typographic attacks via vision-text compression. In contrast to prior work that treats rendered text as adversarial noise, we enhance product retrieval by overlaying relevant textual information directly onto images. This improves image-text alignment in vision-language models without requiring architectural changes or additional model training. Across zero-shot evaluations on proprietary e-commerce datasets and state-of-the-art encoders, our approach consistently improves retrieval performance in both unimodal and multimodal settings. We observe gains in $Acc@1$ and $Acc@3$ across all contrastive learning-based models in our experiment, including the widely used CLIP and SigLIP. Visualizations of attention sensitivity maps and qualitative case studies further support the effectiveness of our approach. Given its minimal latency and resource overhead, the proposed method is well-suited for deployment in real-world multimodal product retrieval systems.

Our study primarily focuses on enhancing multimodal product retrieval in a zero-shot setting using pretrained vision-language models. While preliminary findings suggest that the proposed method may also be beneficial during model fine-tuning and pretraining, we leave a systematic assessment of its generalizability across training paradigms to future work.

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

# Appendix

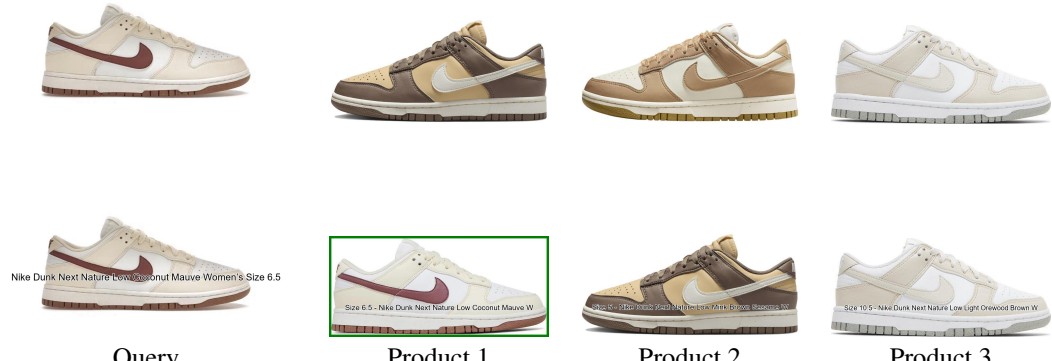

Query   Product 1   Product 2   Product 3

Figure 4: Example from the sneakers category showing the query and retrieved products under raw (top row) and title-rendered (bottom row) conditions. The top row shows the original query image and three retrieved product images. The bottom row shows the corresponding images with listing titles rendered using our proposed method. The correct product match is highlighted with a green box.

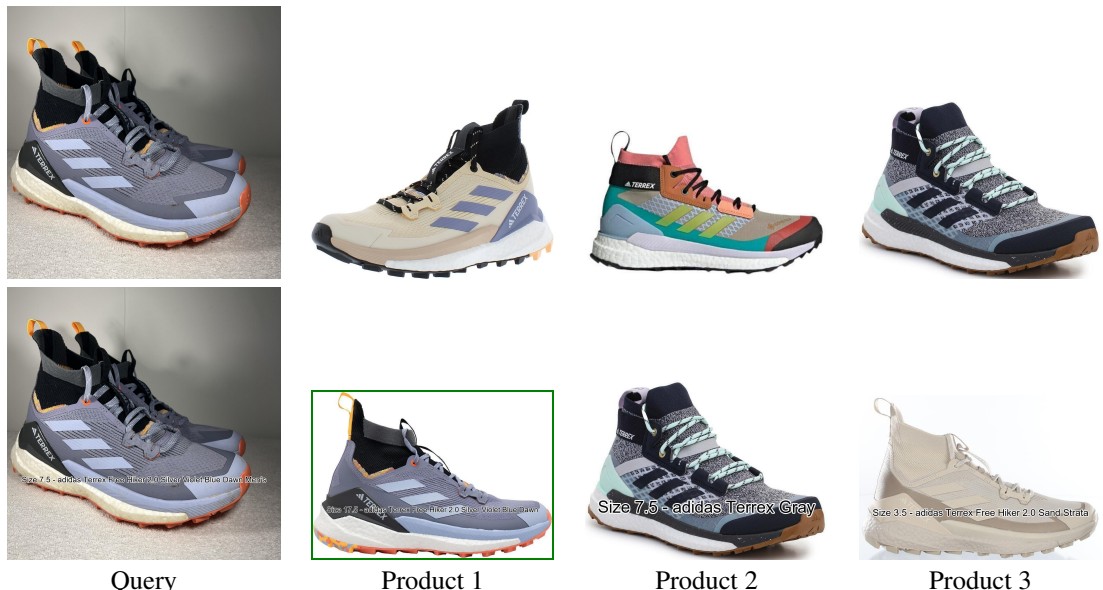

Query   Product 1   Product 2   Product 3

Figure 5: Example from the sneakers category showing the query and retrieved products under raw (top row) and title-rendered (bottom row) conditions. The top row shows the original query image and three retrieved product images. The bottom row shows the corresponding images with listing titles rendered using our proposed method. The correct product match is highlighted with a green box.

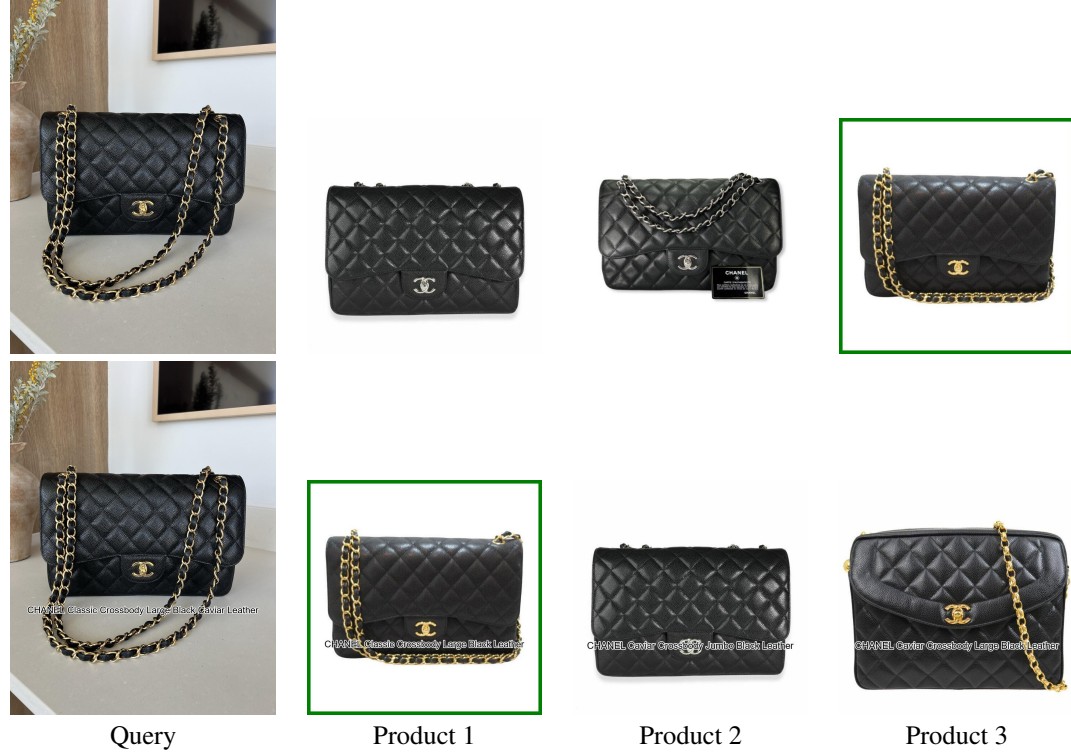

Query          Product 1          Product 2          Product 3

Figure 6: Example from the handbags category showing the query and retrieved products under raw (top row) and title-rendered (bottom row) conditions. The top row shows the original query image and three retrieved product images. The bottom row shows the corresponding images with listing titles rendered using our proposed method. The correct product match is highlighted with a green box.

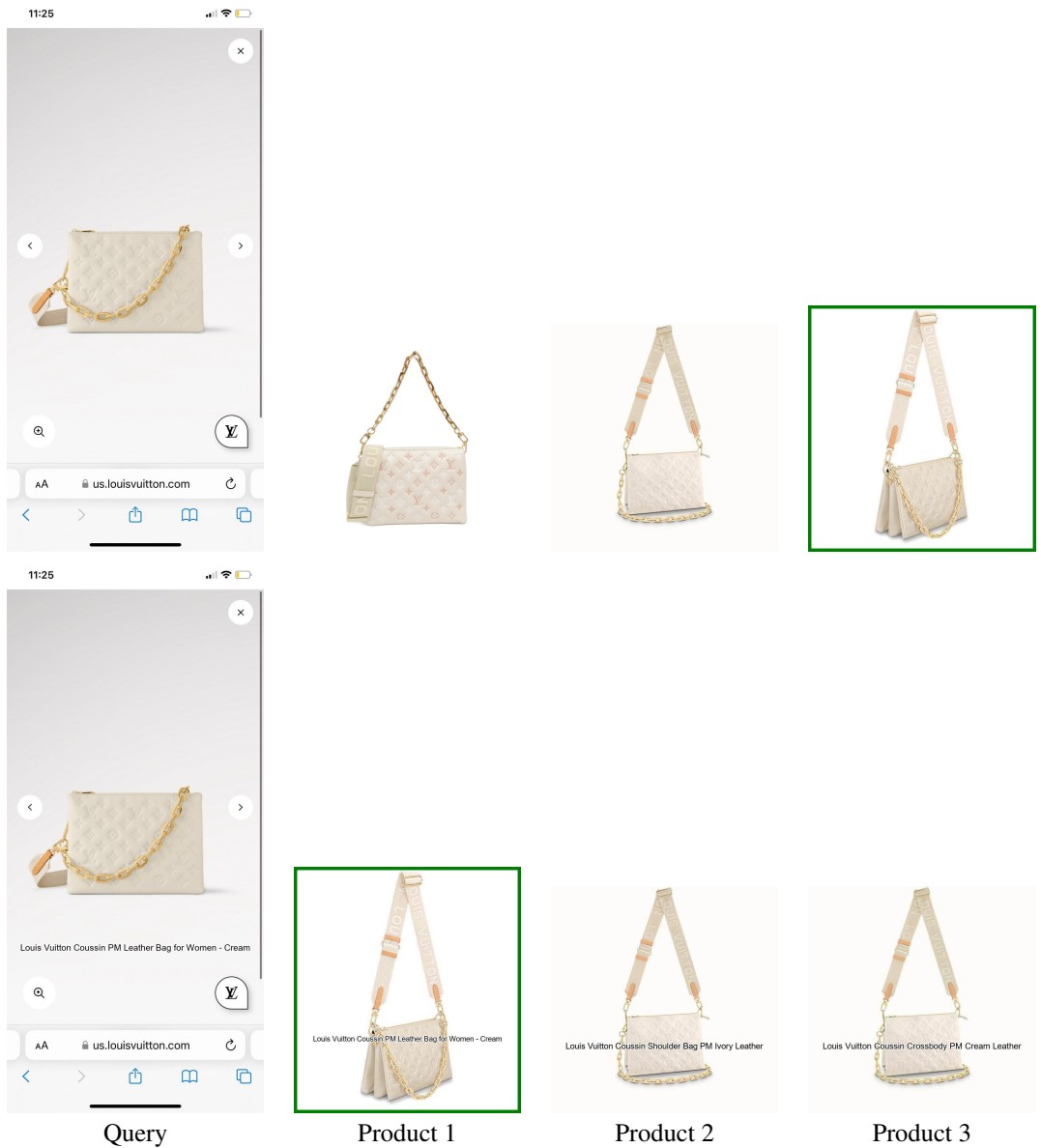

Query          Product 1          Product 2          Product 3

Figure 7: Example from the handbags category showing the query and retrieved products under raw (top row) and title-rendered (bottom row) conditions. The top row shows the original query image and three retrieved product images. The bottom row shows the corresponding images with listing titles rendered using our proposed method. The correct product match is highlighted with a green box.

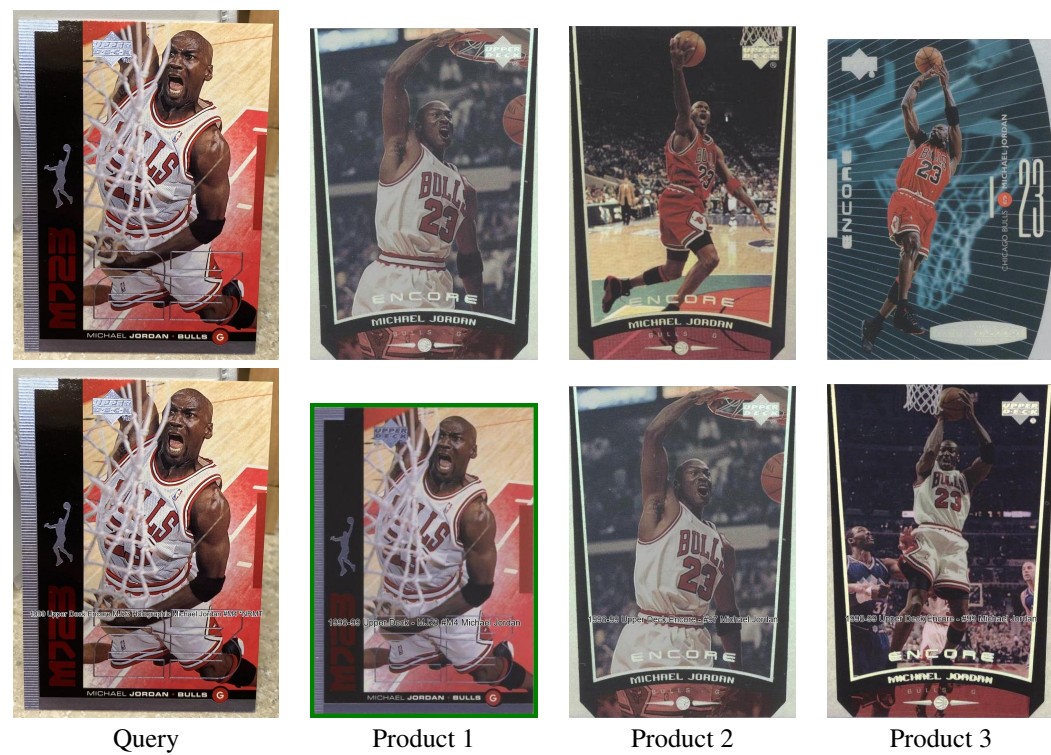

| Query | Product 1 | Product 2 | Product 3 |

Figure 8: Example from the trading cards category showing the query and retrieved products under raw (top row) and title-rendered (bottom row) conditions. The top row shows the original query image and three retrieved product images. The bottom row shows the corresponding images with listing titles rendered using our proposed method. The correct product match is highlighted with a green box.

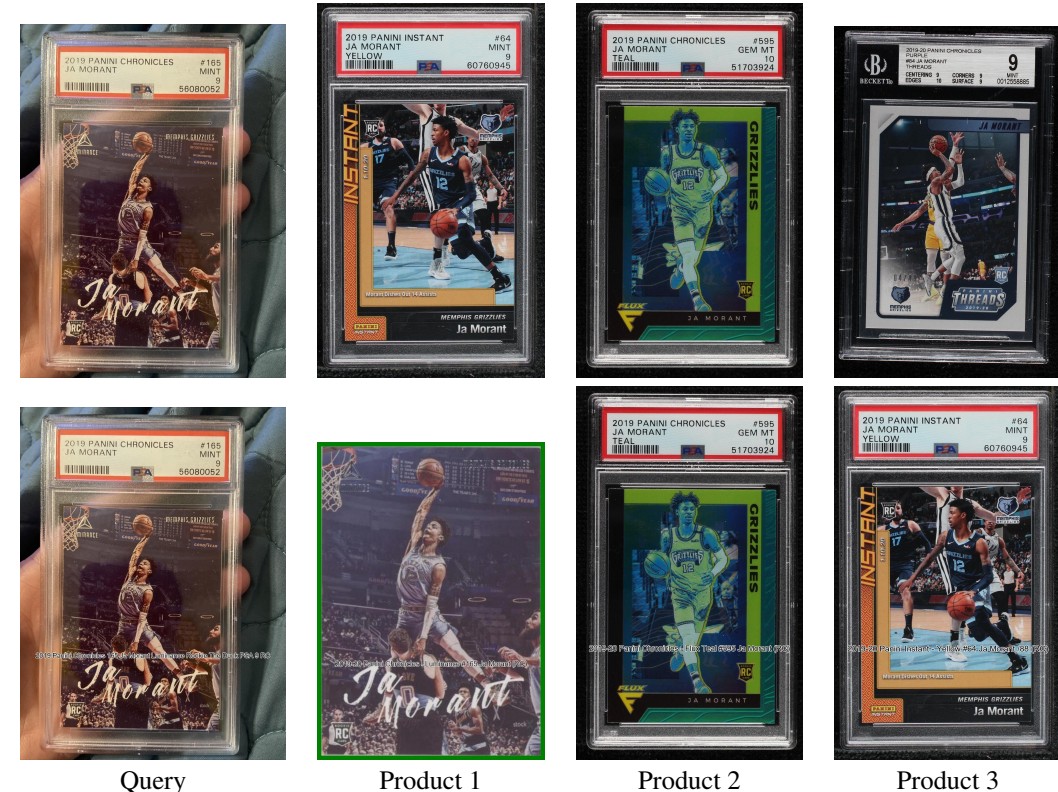

|       |           |           |           |
|-------|-----------|-----------|-----------|
| Query | Product 1 | Product 2 | Product 3 |

Figure 9: Example from the trading cards category showing the query and retrieved products under raw (top row) and title-rendered (bottom row) conditions. The top row shows the original query image and three retrieved product images. The bottom row shows the corresponding images with listing titles rendered using our proposed method. The correct product match is highlighted with a green box.

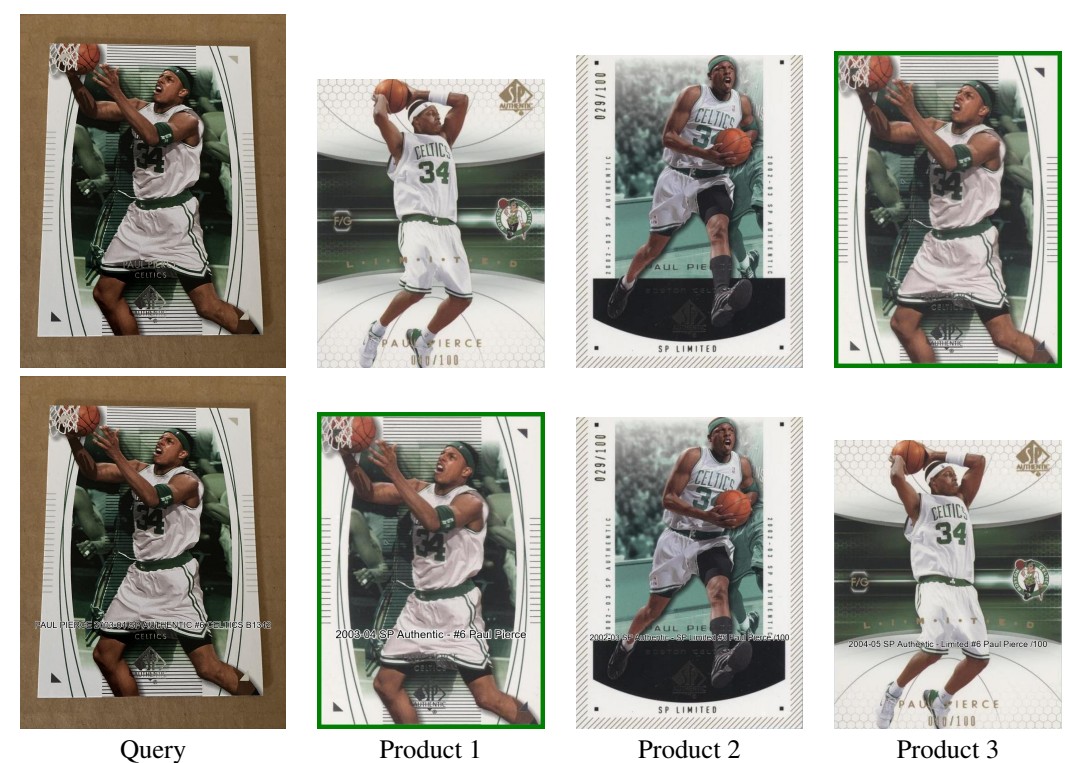

| Query | Product 1 | Product 2 | Product 3 |

Figure 10: Example from the trading cards category showing the query and retrieved products under raw (top row) and title-rendered (bottom row) conditions. The top row shows the original query image and three retrieved product images. The bottom row shows the corresponding images with listing titles rendered using our proposed method. The correct product match is highlighted with a green box.

