# OpenReview forum: "Enhancing Multimodal Product Retrieval in E-Commerce by Reversing Typographic Attacks"
_NeurIPS.cc/2025/Workshop/UniReps — UniReps2025_

### Official Review · Reviewer_4kni · 2025-09-04
**Well defined and inspired contribution!**

**Confidence:** 4

**Review:**

Paper: Enhancing Multimodal Product Retrieval in E-Commerce by Reversing Typographic Attacks

Summary:

The authors propose a novel enhancement of multi-modal product retrieval systems. By embedding relevant text in images they can improve image and text alignment when retrieving relevant products on e-commerce systems. According to the authors, this method takes inspiration from typographic attacks, which does the reverse: Throwing off product retrieval by embedding irrelevant or misaligned information in images.

The method uses two encoding models for image and text encoding respectively. The authors use 3 pipelines for inference. The first of which is a traditional inference fusing encoding results of image and text inputs. The 2nd and 3rd inference modes are proposed as contributions. Inference 2 uses only an image encoding of the image with added relative text, while Inference 3 acts as a combination of 1 and 2: Text is used as an encoded modality (as in inference 1), but is also overlain onto the image prior to image encoding, before both encodings are fused and used to in a product query.

The authors also outline the specific text overlay algorithm used as well, though I cannot argue for or against the novelty of this method.

Results indicate that performance improvements are consistent but depend on the foundation models used to encode and retrieve queries.

Quality and Clarity:

The paper is quite well written. It conveys the methodology and results in an easy to understand way, and uses results to justify their conclusions. The tables and figures are all well organized and convey necessary information. The overall motivation, prior work, and introduction are also well defined and help the authors convey their contributions.

However, highlighting a 3% improvement average seems to distract from conclusive evidence of areas in which improvement was significant. The quantity of data being trained on may or may not result in enough variance to make that metric irrelevant as variance metrics were not provided. This extends to certain improvements when models were performing poorly, such as sub 30% accuracy handbag predictions.

Significance and originality:

Despite my one variance related criticism, this work makes clear the proposed method, which is novel, and shows clear performance increases from state-of-the-art methods. It has a well defined use case and sub-domain to which it would be a valuable contribution!

**Score:**

4

**Topic Fit:**

3

---

### Official Review · Reviewer_MWRE · 2025-09-07
**The paper makes a meaningful contribution that advances the field of multimodal retrieval in e-commerce. While there are areas for improvement, particularly in experimental scope and theoretical understanding, the novel approach, consistent results, and practical applicability outweigh the limitations.**

**Confidence:** 4

**Review:**

## A Brief Overview

This paper proposes a novel approach to improve multimodal product retrieval by overlaying relevant textual information (product titles) directly onto product images, effectively "reversing" typographic attacks. The method strengthens image-text alignment in vision-language models without requiring architectural changes or additional training. The authors evaluate their approach on three e-commerce datasets (sneakers, handbags, trading cards) across five state-of-the-art models (CLIP, PE, SigLIP, SigLIP 2, DINOv2), demonstrating consistent improvements in both unimodal and multimodal retrieval scenarios.

## Strengths

**Novel Conceptual Approach**: The core insight of reversing typographic attacks is creative and well-motivated, transforming a known vulnerability into a beneficial enhancement mechanism.

**Comprehensive Evaluation**: Testing across five different model architectures and three product categories provides strong evidence of generalizability. The inclusion of attention visualization analysis effectively demonstrates the underlying mechanism.

**Consistent Performance Gains**: Results show robust improvements across models, with SigLIP achieving 3.11-point gains in Acc@1 and PE showing 3.00-point improvements, even enhancing already strong baseline models.

**Practical Applicability**: The zero-shot nature, computational efficiency, and minimal integration requirements make the method immediately deployable in production e-commerce systems.

## Weaknesses

**Limited Experimental Scope**:
- Evaluation restricted to proprietary datasets from a single platform
- Small query sets (1,127-2,000 queries) may not represent real-world diversity
- Text modality limited to titles only due to token constraints
- No comparison with other multimodal enhancement techniques

**Insufficient Statistical Rigor**:
- No significance testing or confidence intervals reported
- Single-run evaluation without variance analysis
- Missing analysis of failure cases or method limitations

**Shallow Technical Analysis**:
- Limited exploration of text rendering parameters (font, color, placement)
- SigLIP 2's anomalous behavior inadequately explained
- Binary search font sizing may not optimize for performance vs. legibility trade-offs

**Methodological Concerns**:
- Single human-annotated ground truth may oversimplify evaluation
- No cross-platform or public benchmark evaluation
- Missing comparison with domain-specific e-commerce methods

## Minor Issues

- Algorithm 1 employs fixed design choices without systematic evaluation
- Related work could better position the method within multimodal retrieval literature
- Some qualitative examples could be presented more clearly

## Overall Assessment

This paper presents a relatively solid and creative contribution to multimodal retrieval with clear practical value. The core insight of constructively leveraging typographic sensitivity in vision-language models is novel and well-executed. The comprehensive model evaluation demonstrates broad applicability, and the consistent performance improvements across diverse scenarios provide strong evidence for effectiveness.

However, the work is limited by experimental scope constraints and insufficient statistical rigor. The restriction to proprietary datasets and title-only text reduces broader impact assessment, while the lack of significance testing and variance analysis weakens confidence in the results.

Despite these limitations, the method's immediate practical applicability, zero-shot effectiveness, and theoretical innovation make it a valuable contribution to e-commerce retrieval systems.

The paper should address statistical validation concerns and provide deeper analysis of rendering parameters and failure cases. With these improvements, it would represent a strong contribution to both theoretical understanding and practical application of multimodal retrieval.

**Score:**

4

**Topic Fit:**

2

---

### Official Review · Reviewer_oWTs · 2025-09-16
**Overlaying titles on images improves product retrieval but risks OCR shortcut**

**Confidence:** 4

**Review:**

This paper proposes a lightweight approach for improving multimodal product retrieval in e-commerce by overlaying product titles onto product images. The authors argue that this strategy increases robustness to typographic attacks and evaluate it across three categories: sneakers and handbags (with limited text in images) and trading cards (with more text).

The paper is generally well-written and clearly motivated. The method is described with helpful illustrations. One point that requires clarification is whether text overlays are also applied to the search pool of products $P$. From the description of Inference-B and the figures in the Appendix, it appears that overlays are indeed applied to the gallery images whenever the input query is a rendered image. If so, the reported accuracy gains may largely be explained by direct pixel-level text matching. Since the overlaid text appears in the same font, color, and similar location across images, this substantially simplifies the retrieval task and makes the improvements feel more like an artifact than a genuine advance in multimodal alignment.

From the results, it is clear that adding the product title as a separate text input improves accuracy, but again, this may reduce the problem to something akin to an OCR task when gallery images contain the identical textual overlays. In that case, the improvements primarily reflect the ease of detecting and matching text rather than stronger cross-modal representation learning.

A minor issue: in Table 2 (Handbags, Acc@3, PE), the highest score in bold should be Rendered title concat (0.3645) rather than Rendered title sum (0.3635).

Overall, this paper presents an interesting and practical engineering solution for e-commerce retrieval, but the main improvements seem to come from injecting textual labels onto the visual stream. This risks conflating multimodal alignment with OCR-based text matching. I recommend a weak reject, as the contribution is practical but lacks scientific depth in terms of multimodal represetation.

**Score:**

2

**Topic Fit:**

1

---

### Official Review · Reviewer_VLjf · 2025-09-16
**Well-documented and executed for rendering product titles onto images**

**Confidence:** 5

**Review:**

This is a well-executed, practically motivated study that proposes a simple but effective technique, visually rendering product titles onto images, to improve zero-shot multimodal and unimodal retrieval performance across multiple vision-language encoders and verticals. The idea is clear, original in its inversion of typographic attack logic, and significant for e-commerce search where small relevance gains matter at scale.
The experimental design is sound as it has three distinct product categories with differing levels of inherent visual text (sneakers, handbags, trading cards), five strong vision foundation models spanning contrastive dual-encoders, and a vision-only baseline. This breadth supports the central claim that visually rendering metadata is a generally beneficial enhancement for retrieval. The findings align with recent typographic-attack literature, which shows that models attend to embedded text and leverage that behavior constructively. The paper communicates the motivation and mechanism clearly, reversing typographic attacks by overlaying semantically matched text, and provides a concrete, lightweight rendering algorithm with legibility-aware sizing, which helps reproducibility and interpretability for practitioners.
Recasting typographic attacks as a positive signal injection is a clever adaptation because prior work largely treats embedded text as an adversarial artifact to be mitigated. In contrast, this paper operationalizes it as an alignment amplifier by ensuring the overlaid text matches the ground-truth semantics, which distinguishes it from defense-only or prompt-engineering approaches.
The contribution is practically significant for zero-shot retrieval in commerce as it avoids architectural changes, exploits off-the-shelf encoders, and shows consistent accuracy gains, which can translate to tangible ranking improvements in production search and recommendation systems.

Pros
* Simple, low-overhead enhancement that consistently improves Acc@1/Acc@3 across categories and encoders, including strong modern baselines where marginal gains are hard-won.
* Clear framing that inverts a known vulnerability (typographic attacks) into a beneficial alignment strategy, grounded in recent analyses of attention and text-spotting in VLMs/LMMs.
* Practical inference modes, including a single-encoder, rendered-image-only pathway that can simplify serving and indexing at scale.

Cons
* Reported gains are compelling but would benefit from ablations on rendering choices (font, placement, opacity), robustness to noisy or partially incorrect metadata, and sensitivity to text length constraints, factors known to affect typographic influence.
* The approach may introduce aesthetic clutter or watermark-like artifacts on consumer-facing images; human-perception or UX impacts are not assessed, and production systems may need alternate pathways to avoid user-visible overlays while preserving index-time gains.
* Limited discussion of fairness or safety as over-reliance on embedded text could amplify biases or branding artifacts, a concern raised in broader studies of typographic vulnerabilities and spurious features.

Suggestions
* Include controlled ablations for rendering parameters (font family/size bounds, color/contrast, placement) and OCR occlusion risk, since typographic effects are sensitive to typography and spatial layout.
* Evaluate robustness with noisy/partial/long metadata and compare title vs. curated keyphrase rendering; connect outcomes to prompt-informativeness findings that modulate susceptibility to typographic effects.
* Measure offline latency/throughput effects for rendered-image-only indexing vs. bimodal fusion to quantify practical deployment trade-offs in large catalogs.

Overall assessment: strong accept for practical impact. The work is of high quality, clearly written, and meaningfully original in turning a well-documented weakness into a deployment-friendly enhancement; given consistent zero-shot gains across strong encoders and categories, the method is a notable, actionable contribution for e-commerce retrieval and an interesting data-centric complement to model-centric advances

**Score:**

5

**Topic Fit:**

3